# How to Leverage Imperfect Demonstrations in Offline Imitation Learning

## Abstract

Offline imitation learning (IL) with imperfect data has garnered increasing attention due to the scarcity of expert data in many real-world domains. A fundamental problem in this scenario is *how to extract good behaviors from noisy demonstrations*. In general, current approaches to the problem build upon state-action similarity to the expert, neglecting the valuable information in (potentially abundant) diverse behaviors that deviate from given expert demonstrations. In this paper, we introduce a simple yet effective data selection method that identifies the positive behavior based on its *resultant state*, which is a more informative criterion that enables explicit utilization of dynamics information and the extraction of both expert-like and beneficial diverse behaviors. Further, we devise a lightweight constrained behavior cloning algorithm capable of leveraging the expert and selected data correctly. We term our proposed method *iLID* and evaluate it on a suite of complex and high-dimensional offline IL benchmarks, including MuJoCo and Adroit tasks. The results demonstrate that iLID achieves state-of-the-art performance, significantly outperforming existing methods often by **2-5x** while maintaining a comparable runtime to behavior cloning (BC).

## 1   Introduction

Offline imitation learning (IL) is the study of learning from demonstrated behaviors without reinforcement signals or further interaction with the environment. It has been deemed as a promising solution for safety-sensitive applications, such as autonomous driving and healthcare, where manually identifying a reward function is difficult but historical human demonstrations are readily available. Traditionally, offline IL methods such as behavior cloning (BC) (Pomerleau, 1988) often require an expert dataset with sufficient coverage over state-action spaces to combat error compounding (Ross and Bagnell, 2010; Jarrett et al., 2020; Chan and van der Schaar, 2021), which can be prohibitively expensive for many real-world domains. Instead, a more realistic scenario might allow for a small expert dataset, combined with a large amount of *imperfect data* sampled from unknown policies (Wu et al., 2019; Xu et al., 2022a; Yu et al., 2022). For example, autonomous vehicle companies may have limited high-quality data from experienced drivers but can obtain a wealth of mixed-quality data from ordinary drivers. Clearly, effective utilization of these imperfect demonstrations would significantly enhance the robustness and generalization of offline IL.

A fundamental question raised in this scenario is: *how can we extract good behaviors from noisy data?* To answer this question, several prior works have attempted to explore and imitate the imperfect behaviors that resemble expert ones (as in Xu et al. (2022a); Sasaki and Yamashina (2020)). However, due to the scarcity of expert data, such methods are ill-equipped to leverage valuable knowledge in (potentially abundant) *diverse behaviors* that deviate from limited expert demonstrations. Of course, a natural solution to incorporate these behaviors is inferring a reward function and labeling

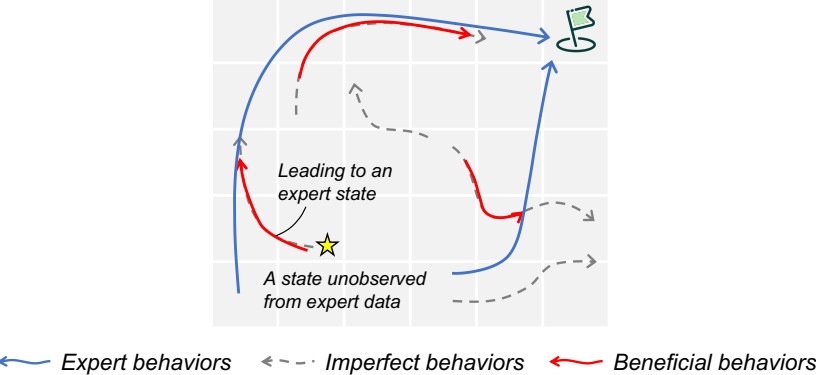

Expert behaviors ←— · · · Imperfect behaviors ←— Beneficial behaviors

Figure 1: A cartoon illustration of the beneficial behaviors in imperfect data.

all imperfect data, followed by an offline reinforcement learning (RL) progress (as in Zolna et al. (2020); Chang et al. (2022); Yue et al. (2023)). Unfortunately, it is highly challenging to define and learn meaningful reward functions without environmental interaction. As a result, current offline reward learning methods typically rely on complex adversarial optimization using a learned dynamics model. They easily suffer from hyperparameter sensitivity, learning instability, and limited scalability in high-dimensional environments (Yu et al., 2022; Arjovsky et al., 2017; Garg et al., 2021).

In this paper, we introduce a simpler data selection method along with a lightweight policy learning algorithm to fully exploit both expert-like and positive diverse behaviors in imperfect demonstrations without indirect reward learning procedures. Specifically, instead of examining a behavior's similarity to expert demonstrations in and of itself, we assess its value based on whether its *resultant state*, to which environment transitions after performing that behavior, falls within the expert data manifold. In other words, we (properly) select the state-actions that can lead to expert states, even if they bear no similarity to expert demonstrations. As illustrated in Fig. 1 and supported by the theoretical results in Section 3.1, the underlying rationale is: when the agent encounters a state unobserved in expert demonstrations, compared to taking a random action, a more reasonable way is to return to the states where it knows expert behaviors; otherwise, it may keep making mistakes and remain out-of-expert-distribution for the remainder of time steps. Notably, the resultant state is more informative than the state-action similarity, as it can explicitly utilize the dynamics information and identify both expert-like and beneficial diverse state-actions.

Drawing on this insight, we first train a *state-only discriminator* to distinguish expert and non-expert states in imperfect demonstrations. Based on the identified expert-like states, we appropriately select their *causal state-actions* and build a complementary training dataset. In light of the suboptimality of the complementary data, we further devise a lightweight constrained BC algorithm to mitigate the potential interference among behaviors. We term our proposed method *Offline Imitation Learning with Imperfect Demonstrations (iLID)* and evaluate it on a suite of offline IL benchmarks, including widely-used MuJoCo tasks as well as more complex and high-dimensional Adroit tasks. iLID achieves state-of-the-art performance, significantly outperforming existing baseline methods often by **2-5x** while maintaining a comparable runtime to BC. In a nutshell, the main contributions of this paper are as follows:

- We introduce a simple yet effective method to select potentially useful behaviors in noisy data. It can explicitly exploit the dynamics information and extract both expert-like and positive diverse behaviors, achieving a significant improvement in the utilization of imperfect demonstrations.

- To avoid behavior interference induced by the suboptimality of complementary behaviors, we propose a constrained BC algorithm that can correctly leverage the expert and extracted behaviors.

- Extensive experiments on complex and high-dimensional domains corroborate that iLID can surpass the existing baseline methods in terms of performance and computational cost.

## 2 Preliminaries

***Episodic Markov decision process (MDP)*** can be specified by $M \doteq \langle \mathcal{S}, \mathcal{A}, T, H, r, \mu \rangle$, consisting of state space $\mathcal{S}$, action space $\mathcal{A}$, transition dynamics $T : \mathcal{S} \times \mathcal{A} \to \mathcal{P}(\mathcal{S})$, episode horizon $H$, reward

function $r : \mathcal{S} \times \mathcal{A} \to [0,1]$, and initial state distribution $\mu : \mathcal{S} \to [0,1]$. A stationary stochastic policy maps states to distributions over actions, denoted as $\pi : \mathcal{S} \to \mathcal{P}(\mathcal{A})$. The policy value of $\pi$ is defined as the expected cumulative reward, $V^\pi \doteq \mathbb{E}[\sum_{h=1}^H r(s_h, a_h)]$, where the expectation is computed w.r.t. the distribution over trajectories induced by rolling out $\pi$ in the environment. The objective of reinforcement learning (RL) can be expressed as $\max_{\pi \in \Pi} V^\pi$, where $\Pi$ is the set of all stationary stochastic policies taking actions in $\mathcal{A}$ given states in $\mathcal{S}$. We denote the average state distribution of policy $\pi$ as $\rho^\pi(s) \doteq \frac{1}{H} \sum_{h=1}^H \Pr(s_h = s | \pi, T, \mu)$, where $\Pr(s_h = s | \pi, T, \mu)$ denotes the probability of visiting $s$ at time step $h$ by rolling out $\pi$ with $M$. When clear from context, we overload notation and denote the average state-action distribution as $\rho^\pi(s, a) \doteq \rho^\pi(s)\pi(a|s)$.

***Offline IL with imperfect demonstrations*** is the setting where the algorithm is neither allowed to interact with the environment nor provided ground-truth reward signals. Rather, it has access to an expert dataset and a mix-quality imperfect/noisy dataset, collected from unknown expert policy $\pi_e$ and (perhaps highly suboptimal) behavior policy $\pi_s$, respectively. To be specific, the expert and imperfect datasets are denoted by $\mathcal{D}_e \doteq \{\tau_j\}_{j=1}^{n_e}$ and $\mathcal{D}_s \doteq \{\tau_i\}_{i=1}^{n_s}$, where $\tau_i \doteq (s_{i,1}, a_{i,1}, \ldots, s_{i,H}, a_{i,H})$ represents a trajectory. Our goal is to learn the best policy with regard to optimizing $V^\pi$ from static offline data $\mathcal{D}_o \doteq \mathcal{D}_e \cup \mathcal{D}_s$ without querying the expert or interacting with the environment.

***Behavior cloning (BC)*** is a classical offline IL approach, which seeks to learn a policy via supervised learning. The standard objective of BC is to maximize the negative log-likelihood over $\mathcal{D}_e$:

$$\max_{\pi \in \Pi} \mathbb{E}_{(s,a) \sim \mathcal{D}_e} \big[ \log(\pi(a|s)) \big]. \tag{1}$$

However, standard BC does not utilize the information in $\mathcal{D}_s$. Due to the limited state coverage of $\mathcal{D}_e$, the learned policy may suffer from severe compounding errors, i.e., the inability for the policy to get back on track if it encounters a state not seen in the expert demonstrations.

# 3   Offline imitation learning with imperfect demonstrations

In this section, we provide a detailed description of our approach. We begin by presenting the theoretical findings on the benefits of utilizing diverse transitions. Building on the theoretical insights, we then design our data selection and policy learning methods.

## 3.1   How to extract good behaviors from noisy data

To discard low-quality demonstrations from $\mathcal{D}_s$, existing approaches often rely on the state-action dissimilarity between $\mathcal{D}_s$ and $\mathcal{D}_e$. For example, Xu et al. (2022a); Zolna et al. (2020); Kim et al. (2022) propose to learn a weighting function $f(s,a)$ by pushing up its value on $(s,a) \in \mathcal{D}_e$ while pushing down that on $(s,a) \in \mathcal{D}_s$. Based on $f(s,a)$, they perform weighed BC to implicitly select expert-like state-actions, i.e., $\max_{\pi \in \Pi} \mathbb{E}_{(s,a) \sim \mathcal{D}_o}[f(s,a)\log(\pi(a|s))]$. However, due to the limitation of expert demonstrations, the learned $f(s,a)$ can be overly conservative and neglect the useful information in diverse state-actions. Therefore, it calls for a more informative criterion to assess the value of imperfect behaviors.

Before preceding, we first provide the following theoretical results under deterministic transition dynamics to gain insights into this problem. Denote $\mathcal{S}_h(\mathcal{D})$ as the set of $h$-step visited states of $\mathcal{D}$ and $\mathcal{S}(\mathcal{D}) \doteq \bigcup_{h=1}^H \mathcal{S}_h(\mathcal{D})$ all the states thereof. Assume that $\pi_e$ is optimal and deterministic, and there exists a supplementary dataset consisting of transition tuples from initial states to given expert states, i.e., $\tilde{\mathcal{D}} \doteq \{(s_i, a_i, s_i') \mid s_i \sim \mu, s_i' \sim \mathcal{S}(\mathcal{D}_e), T(s_i, a_i) = s_i', i = 1, 2, \ldots, \tilde{n}\}$. Consider a policy $\tilde{\pi}$ such that in expert states $\mathcal{S}(\mathcal{D}_e)$, it takes the corresponding expert actions, and in states $\mathcal{S}_1(\tilde{\mathcal{D}}) \backslash \mathcal{S}_1(\mathcal{D}_e)$, it takes the actions in $\tilde{\mathcal{D}}$, that is,

$$\tilde{\pi}(a|s) \doteq \begin{cases} \frac{\sum_{(\tilde{s}, \tilde{a}) \in \mathcal{D}_e} \mathbb{1}((\tilde{s}, \tilde{a}) = (s, a))}{\sum_{\tilde{s} \in \mathcal{S}(\mathcal{D}_e)} \mathbb{1}(\tilde{s} = s)}, & \text{if } s \in \mathcal{S}(\mathcal{D}_e); \\ \frac{\sum_{(\tilde{s}, \tilde{a}) \in \tilde{\mathcal{D}}} \mathbb{1}((\tilde{s}, \tilde{a}) = (s, a))}{\sum_{\tilde{s} \in \mathcal{S}(\tilde{\mathcal{D}})} \mathbb{1}(\tilde{s} = s)}, & \text{if } s \in \mathcal{S}_1(\tilde{\mathcal{D}}) \backslash \mathcal{S}_1(\mathcal{D}_e); \\ \frac{1}{|\mathcal{A}|}, & \text{else.} \end{cases} \tag{2}$$

We bound the suboptimality gap and sample complexity of $\tilde{\pi}$ in the next theorem and corollary.

**Theorem 3.1.** *For any finite and episodic MDP with deterministic transition dynamics and $\mu = U(\mathcal{S})$, the following fact holds:*

$$V^{\pi_e} - \mathbb{E}\big[V^{\tilde{\pi}}\big] \leq \left(\frac{1+\delta}{2} + \frac{1-\delta}{H^2}\right)H\epsilon, \tag{3}$$

*where $\epsilon \doteq \mathbb{E}[\mathbb{E}_{s_1 \sim \mu}[\mathbb{1}(s_1 \notin \mathcal{S}_1(\mathcal{D}_e))]]$ and $\delta \doteq \mathbb{E}[\mathbb{E}_{s_1 \sim \mu}[\mathbb{1}(s_1 \notin \mathcal{S}_1(\tilde{\mathcal{D}}))]]$ represent the missing mass over the initial distribution w.r.t. $\mathcal{S}_1(\mathcal{D}_e)$ and $\mathcal{S}_1(\tilde{\mathcal{D}})$). $U(\mathcal{S})$ is the uniform distribution over $\mathcal{S}$.*

*Proof Sketch.* Note that the error stems from the initial states that are not covered by $\mathcal{S}_1(\mathcal{D}_e)$. We bound the errors generated from the states not in $\mathcal{S}(\mathcal{D}_e) \cup \mathcal{S}(\tilde{\mathcal{D}})$ and from the states in $\mathcal{S}(\tilde{\mathcal{D}}) \backslash \mathcal{S}(\mathcal{D}_e)$ by $H\delta\epsilon$ and $(H/2 + 1/H)(1-\delta)\epsilon$, respectively. Combining these two errors yields the result. For a detailed proof, please refer to Appendix B. $\qquad \square$

Building on Theorem 3.1, we can obtain the following sample complexity result (where we retain the constant $\frac{1}{2}$ in the asymptotic result to highlight the improvement over BC).

**Corollary 3.2.** *Suppose $\tilde{\mathcal{D}}$ is sufficiently large. For any finite and episodic MDP with deterministic transition dynamics and $\mu = U(\mathcal{S})$, to obtain an $\varepsilon$-optimal policy, $V^{\pi_e} - \mathbb{E}[V^{\tilde{\pi}}] \leq \varepsilon$, $\tilde{\pi}$ requires at most $\mathcal{O}(|\mathcal{S}|H/(2 \cdot \varepsilon))$ expert trajectories.*

*Proof.* Invoking Xu et al. (2021, Theorem 2) yields the bounds for the missing mass:

$$\mathbb{E}\left[\mathbb{E}_{s_1 \sim \mu}\left[\mathbb{1}(s_1 \notin \mathcal{S}_1(\tilde{\mathcal{D}}))\right]\right] \leq \frac{|\mathcal{S}|}{e|\tilde{\mathcal{D}}|}, \quad \mathbb{E}\left[\mathbb{E}_{s_1 \sim \mu}\left[\mathbb{1}(s_1 \notin \mathcal{S}_1(\mathcal{D}_e))\right]\right] \leq \frac{|\mathcal{S}|}{e|\mathcal{D}_e|},$$

where $e$ is the Euler's number. If $\tilde{D}$ is sufficiently large, then $\delta \to 0$. Using Theorem 3.1, the result can be easily derived. $\qquad \square$

***Remarks.*** It is worth noting that the minimax suboptimality of BC is limited to $H\epsilon$ in this setting (Rajaraman et al., 2020), and beating the $\mathcal{O}(H)$ barrier is unattainable. The reason is that when the agent encounters a state beyond given demonstrations during the interaction with the environment, it has no prior knowledge about the expert. As a result, the agent is essentially forced to take an arbitrary action in these states, potentially leading to mistakes for $H$ time steps. Whereas, as revealed by Theorem 3.1 and Corollary 3.2, $\tilde{\pi}$ provably alleviates the error compounding and reduces the sample complexity bound of BC (which is $\mathcal{O}(|S|H/\varepsilon)$) by approximately half. The reason behind is that $\tilde{\mathcal{D}}$ can empower $\tilde{\pi}$ to recover from mistakes. Combined with Eq. (2), this provides an important insight for us: in the states uncovered by $\mathcal{D}_e$, if an action can lead to known expert states, mimicking it can benefit the performance of imitation policy.

Thus motivated, we propose to assess the imperfect behavior based on its resultant states rather than the state-action in and of itself. For example, if there exists $(s_1, a_1, s_2, a_2, s_3) \in \mathcal{D}_s$ such that $s' \in \mathcal{D}_e$, one can select $(s_1, a_1)$ and $(s_2, a_2)$ (or only $(s_2, a_2)$), even if these behaviors do not bear similarity to any $(s, a) \in \mathcal{D}_e$. To this end, we consider learning a state-only discriminator to contrast expert and non-expert states in $\mathcal{D}_s$, e.g.,

$$\max_d \mathbb{E}_{s \sim \mathcal{D}_e}\big[\log d(s)\big] + \mathbb{E}_{s \sim \mathcal{D}_s}\big[\log(1 - d(s))\big]. \tag{4}$$

However, optimizing Problem (4) can lead to the problem of *false negative*, where the learned discriminator assigns 1 to all transitions from $\mathcal{D}_e$ and 0 to all transitions from $\mathcal{D}_s$. This problem is analogous to the positive-unlabeled (PU) classification problem (Elkan and Noto, 2008), where both positive (expert) and negative (imperfect) samples exist in the unlabeled data (imperfect demonstrations). Akin to Xu et al. (2022a); Zolna et al. (2020), we adopt the reweighting method from PU learning to address this issue:

$$d^* = \arg\max_d \eta \cdot \mathbb{E}_{s \sim \mathcal{D}_e}\big[\log d(s)\big] + \mathbb{E}_{s \sim \mathcal{D}_s}\big[\log(1 - d(s))\big] - \eta \cdot \mathbb{E}_{s \sim \mathcal{D}_e}\big[\log(1 - d(s))\big], \tag{5}$$

where $\eta > 0$ is a reweighting parameter, corresponding to the proportion of expert states to imperfect states. Intuitively, the third term in Eq. (5) could avoid $d^*(s)$ of the states from $\mathcal{S}(\mathcal{D}_s)$ but similar to $\mathcal{S}(\mathcal{D}_e)$ becoming 0.

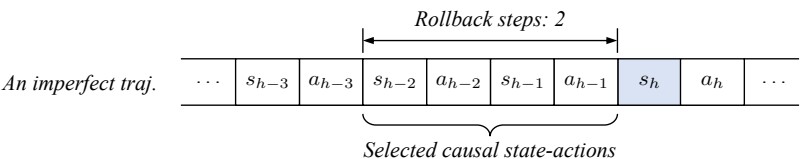

Figure 2: An illustration of the data selection procedure. $s_h$ represents an identified expert state.

**Data selection.** The learned discriminator $d^*$ is able to identify the expert states in $\mathcal{D}_s$. Based on these states, we in turn select their *causal states and actions* to construct a complementary dataset $\tilde{\mathcal{D}}$. Specifically, given threshold $\sigma \in [0,1]$ and *rollback* steps $K \geq 1$, if there exist $h > 1$ and $i \in \{1, \ldots, n_s\}$ (where $n_s$ represents the number of trajectories in $\mathcal{D}_s$) such that $d(s_{i,h}) \geq \sigma$, we include $K$ causal state-action pairs from $s_{i,h}$ into $\tilde{\mathcal{D}}$.:

$$\tilde{\mathcal{D}} \leftarrow \tilde{\mathcal{D}} \cup \left\{ (k, s_{i,h-k}, a_{i,h-k}) \right\}_{k=1:\min\{h-1,K\}} . \tag{6}$$

We iterate the above process for all identified expert-like states. To clarify, we illustrate the process in Fig. 2. It is evident that $\tilde{\mathcal{D}}$ comprises of both the positive diverse state-actions in $\mathcal{D}_s$ and those similar to $\mathcal{D}_e$ therein. This highlights that using resultant states is a more informative way to extract useful behaviors.

**Behavior interference.** After obtaining $\tilde{\mathcal{D}}$, a natural solution to learn an imitation policy is carrying out BC from the union of $\mathcal{D}_e$ and $\tilde{\mathcal{D}}$. However, due to the suboptimality of $\tilde{\mathcal{D}}$, this naïve solution will suffer from potential interference among behaviors. That is, for a selected $(s, a, s')$, if $s, s' \in \mathcal{D}_e$ but $a \neq \pi_e(s)$, action $a$ will affect mimicking the expert behavior in expert state $s$ when learning via the naïve solution. Furthermore, this interference issue also exists in the states of complementary dataset $\tilde{\mathcal{D}}$, but in a more subtle manner. Consider a state $s \in \tilde{\mathcal{D}}$ where two actions $a_1, a_2$ are selected, i.e., $(k_1, s, a_1), (k_2, s, a_2) \in \tilde{\mathcal{D}}$. Owing to the stochasticity of MDPs, if $k_1 < k_2$, one may prefer $a_1$ over $a_2$, whereas the naïve solution will imitate both actions equally. In Section 3.2, we address this problem and propose a lightweight algorithm to correctly learn from $\mathcal{D}_e$ and $\tilde{\mathcal{D}}$.

## 3.2 How to learn an imitation policy from expert and extracted data

Due to the suboptimality of $\mathcal{D}_s$ and the stochasticity of MDPs, direct cloning the behaviors in $\tilde{\mathcal{D}} \cup \mathcal{D}_e$ can lead to the interference issue. In fact, the solution has been implied in Eq. (2), which suggests that the policy should be constrained to $\mathcal{D}_e(\cdot|s)$ in the known expert states. Accordingly, we cast the problem of learning policy from $\mathcal{D}_e$ and $\tilde{\mathcal{D}}$ as follows:

$$\min_{\pi \in \Pi} \mathbb{E}_{(k,s,a) \sim \tilde{\mathcal{D}}} \left[ -\gamma^k \log \pi(a|s) \right] \quad \text{s.t. } \mathbb{E}_{s \sim \mathcal{D}_e} \left[ D_{\mathrm{KL}}(\tilde{\pi}_e(\cdot|s) \| \pi(\cdot|s)) \right] < \epsilon \tag{7}$$

where $\tilde{\pi}_e = \arg\max_{\pi \in \Pi} \mathbb{E}_{(s,a) \sim \mathcal{D}_e}[\log(\pi(a|s))]$ is the BC policy learned on $\mathcal{D}_e$, and $\epsilon \geq 0$ is the threshold. In Eq. (7), we use discount factor $\gamma \in (0, 1]$ to mitigate the impact of stochasticity of MDPs. It is easy to see that with a sufficiently small $\epsilon$, the optimal solution of Problem (7) enjoys at least the same theoretical guarantee of BC in general stochastic MDPs, i.e., suboptimality upper-bound $\mathcal{O}(|\mathcal{S}|H^2 \log n_e/n_e)$ compared to $V^{\pi_e}$ (Rajaraman et al., 2020).

Problem (7) is a convex optimization problem. From Slater's condition, the strong duality holds, and thus the optimization is equal to

$$\max_{\alpha > 0} \min_{\pi \in \Pi} -\mathbb{E}_{k,s,a \sim \tilde{\mathcal{D}}} \left[ \gamma^k \log \pi(a|s) \right] - \alpha \left( \mathbb{E}_{s,a \sim \mathcal{D}_e} \left[ \log \pi(a|s) \right] + \tilde{H} + \epsilon \right), \tag{8}$$

where $\alpha$ is the dual variable, and $\tilde{H}$ is the expected entropy of the empirical expert policy, which is derived from:

$$\mathbb{E}_{s \sim \mathcal{D}_e} \left[ D_{\mathrm{KL}}(\tilde{\pi}_e(\cdot|s) \| \pi(\cdot|s)) \right] = \mathbb{E}_{s \sim \mathcal{D}_e} \left[ \mathbb{E}_{a \sim \tilde{\pi}_e(\cdot|s)} \left[ \log \tilde{\pi}_e(a|s) \right] - \mathbb{E}_{a \sim \tilde{\pi}_e(\cdot|s)} \left[ \log \pi(a|s) \right] \right]$$
$$= \underbrace{\mathbb{E}_{(s,a) \sim \mathcal{D}_e} \left[ \log \tilde{\pi}_e(a|s) \right]}_{\doteq -\tilde{H}} - \mathbb{E}_{(s,a) \sim \mathcal{D}_e} \left[ \log \pi(a|s) \right] . \tag{9}$$

---

**Algorithm 1:** Offline Imitation Learning with Imperfect Demonstrations (iLID)

**Require:** expert data $\mathcal{D}_e$, imperfect data $\mathcal{D}_s$, learning rate $\lambda$, parameter $K, \epsilon$

1 Train discriminator $d$ using $\mathcal{D}_e$ and $\mathcal{D}_s$ based on Eq. (5);
2 Select data from $\mathcal{D}_s$ and build complementary dataset $\tilde{\mathcal{D}}$ based on Eq. (6);
3 Train BC policy $\tilde{\pi}_e$ only using $\mathcal{D}_e$ and compute expected entropy $\tilde{H} \doteq -\mathbb{E}_{s,a\sim\mathcal{D}_e}[\log \tilde{\pi}_e(a|s)]$;
4 Initialize policy $\pi_\theta$ and dual variable $\alpha$;
5 **while** *not done* **do**
6     Sample a training batch from $\mathcal{D}_e$ and $\tilde{\mathcal{D}}$;
7     Update policy parameter $\theta \leftarrow \theta - \lambda\tilde{\nabla}L(\theta)$ based on Eq. (10);
8     Update dual variable $\alpha \leftarrow \alpha - \lambda\tilde{\nabla}L(\alpha)$ based on Eq. (11);
9 **end while**

---

Parameterize the learned policy by $\theta$ and denote the loss functions for $\theta$ and $\alpha$ as follows:

$$L(\theta) \doteq -\mathbb{E}_{k,s,a\sim\tilde{\mathcal{D}}}\big[\gamma^k \log \pi_\theta(a|s)\big] - \alpha\mathbb{E}_{s,a\sim\mathcal{D}_e}\big[\log \pi_\theta(a|s)\big], \tag{10}$$

$$L(\alpha) \doteq \mathbb{E}_{s,a\sim\mathcal{D}_e}\big[\log \pi_\theta(a|s)\big] + \tilde{H} + \epsilon. \tag{11}$$

Problem (8) can be optimized by *approximating dual gradient descent* that alternates between the gradient steps w.r.t. $L(\theta)$ and $L(\alpha)$, which has been shown to converge under convexity assumptions (Boyd and Vandenberghe, 2004) and work very well in the case of nonlinear function approximators such as neural networks (Haarnoja et al., 2018).

Our algorithm, named *Offline Imitation Learning with Imperfect Demonstrations (iLID)*, is outlined in Algorithm 1. Notably, while iLID pretrains a discriminator and a BC policy (using $\mathcal{D}_e$), the progress can converge within a small number of gradient steps, especially when $\mathcal{D}_e$ is limited. In light of the negligible cost in updating $\alpha$, iLID is indeed computationally cheap.

## 4 Experiments

In this section, we use experimental studies to test the proposed method and answer the following questions: *1) Can iLID effectively utilize imperfect demonstrations? 2) What is the convergence property of iLID? 3) How does iLID perform given different numbers of expert demonstrations or different qualities of imperfect demonstrations? 4) What is the impact of the rollback steps? 5) What is the runtime of iLID? 6) Is the constrained BC an overkill?*

**Baselines.** We evaluate our method against five strong baseline methods in the offline IL setting: *1) Behavior Cloning with Expert Data (BCE)*, the standard BC trained only on the expert dataset (Pomerleau, 1988); *2) Behavior Cloning with Union Data (BCU)*, BC on both the expert and diverse datasets; *3) Discriminator-Weighted Behavioral Cloning (DWBC)* (Chang et al., 2022), a recent offline IL algorithm capable of leveraging suboptimal demonstrations; *4) Using Imperfect Demonstration via Stationary Distribution Correction Estimation (DemoDICE)* (Kim et al., 2022), another recent offline IL algorithm that can leverage suboptimal demonstrations; *5) Conservative offLine model-bAsed Reward lEarning (CLARE)* (Yue et al., 2023), a recent model-based offline inverse RL algorithm trained from both expert and imperfect datasets.

**Datasets.** We conduct experiments on both widely-used MuJoCo tasks (including HalfCheetah, Walker2d, Hopper, and Ant) and more complex and high-dimensional Adroit tasks (including Pen, Hammer, Relocate, and Door, shown on the right). We use the D4RL datasets (Fu et al., 2020) and utilize the `random` and `expert` data for each MuJoCo task, and `cloned` and `expert` data for Adroit tasks.[1] Similar to Xu et al. (2022a); Kim et al. (2022), we generate $\mathcal{D}_e$ and $\mathcal{D}_s$ as follows:

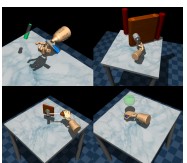

---

[1]Experimental details is elaborated in Appendix A.

Table 1: Performance of different algorithms. The numbers of expert trajectories are 1 for MuJoCo tasks and 10 for Adroit. The results correspond to the mean and standard deviation of normalized scores over 5 random seeds. *low* and *high* represent the qualities of imperfect data.

| Task | Data quality | BCE | BCU | DWBC | ClARE | DemoDICE | iLID (ours) |
|---|---|---|---|---|---|---|---|
| Ant | *low* | -11.1 ± 9.7 | 31.4 ± 0.1 | 30.6 ± 9.7 | 29.7 ± 6.4 | 74.3 ± 11.0 | **79.8 ± 11.8** |
| | *high* | -11.1 ± 9.7 | 32.4 ± 7.1 | 34.6 ± 8.7 | 22.4 ± 4.7 | 88.1 ± 8.9 | **88.2 ± 7.9** |
| HalfCheetah | *low* | 0.2 ± 0.9 | 2.2 ± 0.0 | 1.1 ± 1.1 | 1.1 ± 0.9 | 2.2 ± 0.0 | **25.4 ± 4.1** |
| | *high* | 0.2 ± 0.9 | 2.3 ± 0.0 | 0.8 ± 1.2 | 2.2 ± 0.9 | 5.9 ± 2.8 | **29.3 ± 6.3** |
| Hopper | *low* | 17.0 ± 4.2 | 7.6 ± 5.7 | 76.0 ± 9.4 | 8.9 ± 5.2 | 58.3 ± 13.8 | **95.0 ± 10.9** |
| | *high* | 17.0 ± 4.2 | 3.7 ± 1.6 | 60.6 ± 18.6 | 3.5 ± 0.5 | 72.2 ± 13.6 | **104.8 ± 7.1** |
| Walker2d | *low* | 8.0 ± 5.7 | 0.3 ± 0.1 | 61.1 ± 13.9 | 1.9 ± 0.8 | 96.7 ± 7.5 | **97.0 ± 8.0** |
| | *high* | 8.0 ± 5.7 | 0.3 ± 0.0 | 49.9 ± 26.5 | 1.4 ± 0.5 | **102.6 ± 6.3** | 97.0 ± 10.3 |
| Hammer | *low* | 6.8 ± 5.6 | 0.2 ± 0.0 | 11.0 ± 8.8 | 7.2 ± 8.3 | 10.1 ± 12.3 | **66.0 ± 17.8** |
| | *high* | 6.8 ± 5.6 | 0.2 ± 0.0 | 13.2 ± 7.1 | 3.9 ± 4.4 | 9.1 ± 12.5 | **109.4 ± 10.0** |
| Pen | *low* | -0.1 ± 0.0 | 2.1 ± 6.9 | 43.7 ± 14.2 | 7.5 ± 5.9 | 41.3 ± 13.9 | **90.2 ± 19.4** |
| | *high* | -0.1 ± 0.0 | 1.6 ± 3.4 | 57.1 ± 13.6 | 6.4 ± 6.6 | 48.6 ± 25.3 | **65.7 ± 7.5** |
| Relocate | *low* | -0.1 ± 0.0 | -0.1 ± 0.0 | -0.1 ± 0.0 | 0.0 ± 0.0 | 12.0 ± 5.6 | **29.1 ± 5.6** |
| | *high* | -0.1 ± 0.0 | 0.0 ± 0.0 | -0.1 ± 0.1 | 0.0 ± 0.0 | 26.0 ± 10.6 | **41.5 ± 12.1** |
| Door | *low* | **1.0 ± 1.2** | -0.1 ± 0.0 | 0.5 ± 1.0 | -0.1 ± 0.0 | -0.1 ± 0.1 | 0.3 ± 0.4 |
| | *high* | **1.0 ± 1.2** | -0.1 ± 0.1 | 0.3 ± 0.7 | -0.2 ± 0.1 | -0.1 ± 0.0 | 0.4 ± 0.4 |

- *Expert datasets:* For MuJoCo, we sample 1 trajectory (including less than 1000 state-action pairs) from the `expert` D4RL dataset to constitute expert datasets. For Adroit, we sample 10 `expert` trajectories (each includes less than 100 state-action pairs) to form the datasets.

- *Imperfect datasets:* For MuJoCo tasks, we sample 1000 `random` trajectories mixed with 10 and 20 `expert` trajectories to constitute the low-quality and high-quality imperfect datasets. Regarding Adroit tasks, we sample 1000 `cloned` trajectories mixed with 100 and 200 `expert` trajectories to constitute the low-quality and high-quality datasets.

**Comparative results.** To answer the first and second questions, we show the comparative results under both low-quality and high-quality imperfect data in Section 4 and their corresponding learning curves in Fig. 4. iLID outperforms baseline algorithms on most of the tasks (13 out of 16) often by a wide margin and reaches near-expert scores on many tasks. It indicates that iLID can effectively extract and leverage positive behaviors from imperfect demonstrations over the approaches based on state-action similarity such as DWBC and DemoDICE. Unsurprisingly, BCE fails to fulfill most of the tasks, while BCU learns a mediocre policy. CLARE also performs poorly because the learned reward function could become too pessimistic due to the scarcity of expert demonstrations. Clearly, the model-based approach struggle in high-dimensional environments.

**Expert demonstrations.** To answer the third question, we vary the numbers of expert trajectories from 1 to 50 and present the results on Fig. 3(a). iLID reaches the expert with sufficient expert data.

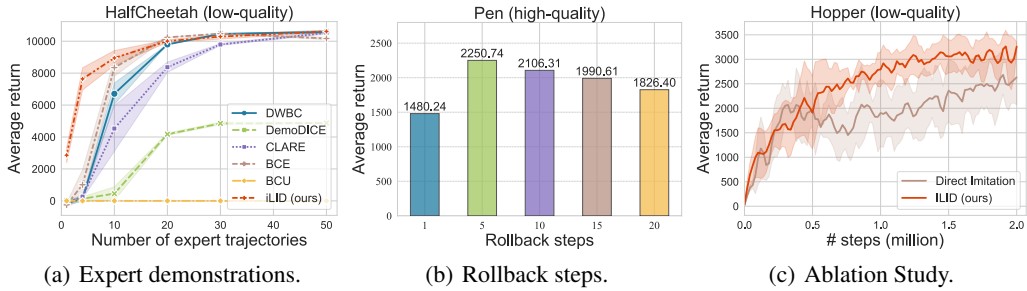

(a) Expert demonstrations.  (b) Rollback steps.  (c) Ablation Study.

Figure 3: Performance of iLID under varying numbers of expert demonstrations and rollback steps along with the ablation study for the constrained BC procedure.

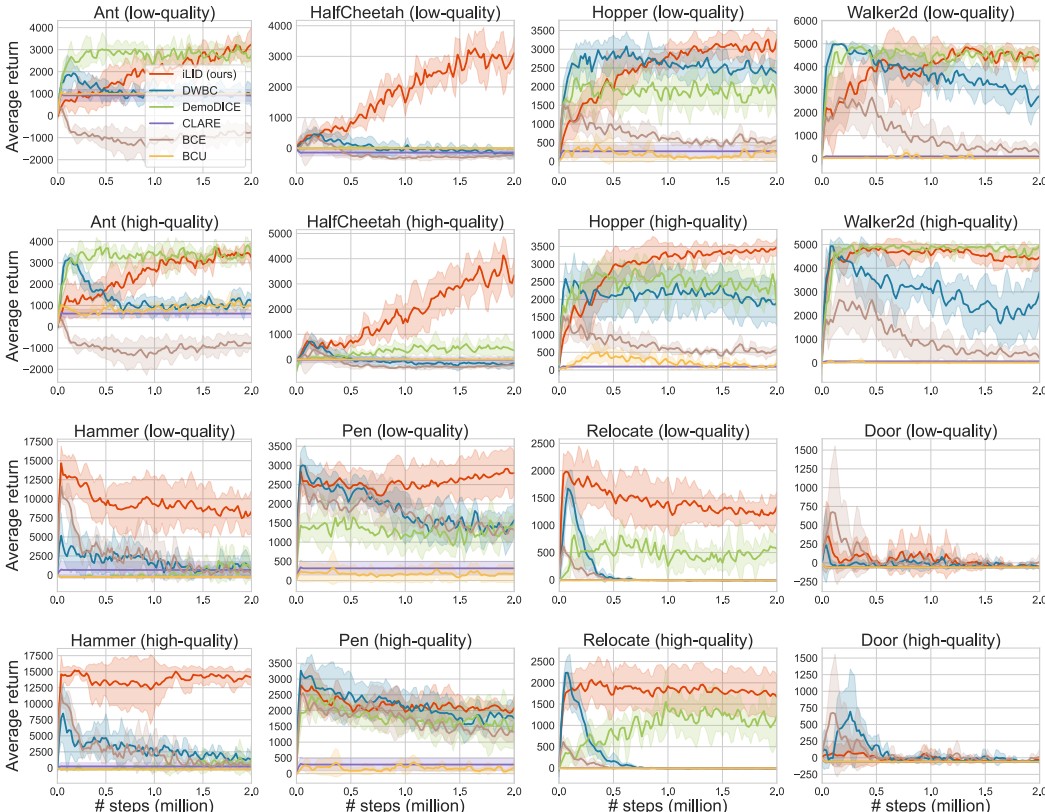

Figure 4: Convergence properties of different algorithms. The solid curve corresponds to the mean and the shaded region to the standard derivative across five random seeds.

Albeit with very limited expert trajectories, iLID also achieves strong performance, revealing its advantages in extracting good behaviors. DemoDICE performs relatively poorly with larger $n_e$. The reason is that it learns on both `expert` and `random` data, whereas the `random` data of HalfCheetah is highly suboptimal.

**Rollback steps.** To answer the fourth question, we vary the rollback steps from 1 to 20 and show the corresponding results in Fig. 3(b). With larger $K$, the performance increases at the beginning. This is due to more positive diverse data included. An excessively large $K$ may have a negative impact due to the dynamics stochasticity and behavior interference. However, it is worth noting that, as $K$ increases further, the performance does not significantly deteriorate. This is because we apply a discount factor to penalize the potential uncertainty in the resulting states, capable of mitigating the issue. In practice, $K$ can be treated as a hyper-parameter to tune. Intuitively, it can be set relatively smaller in a more stochastic environment.

**Ablation study.** We compare iLID to the naïve solution mentioned in Section 3.1, i.e., directly imitating the union of expert and select data. Fig. 3(c) validates the necessity of the constrained BC procedure. The result of *direct imitation* is passable as we select a number of positive data. However, it fails to deal with the behavior interference issue caused by the suboptimality of imperfect data.

**Runtime.** We evaluate the runtime of iLID compared with baseline algorithms for 250,000 training steps, utilizing the same network size and batch size. We reproduce the reported results in Xu et al. (2022a) on an NVIDIA V100 GPU. As illustrated by the figure on the right, the runtime of iLID is nearly the same as BC. It substantiates that the iLID is indeed a lightweight method. Due to the cooperation training between the discriminator and policy, DWBC requires additional computation than iLID. CLARE is costly due to the effort to solve an intermediate offline RL problem.

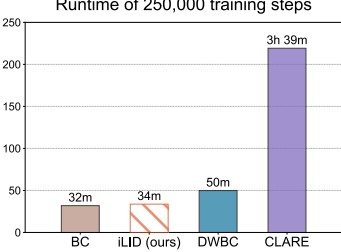

## 5 Related work

Offline IL deals with training an agent to mimic the actions of a demonstrator in an entirely offline fashion. BC (Ross and Bagnell, 2010) is an intrinsically offline solution, but it is prone to covariate shift and inevitably suffers from error compounding, i.e., there is no way for the policy to learn how to recover if it deviates from the expert behavior to a state not seen in the expert demonstrations (Levine et al., 2020). Considerable research has been devoted to developing new offline IL methods to remedy this problem, e.g., Jarrett et al. (2020); Chan and van der Schaar (2021); Garg et al. (2021); Klein et al. (2011, 2012); Piot et al. (2014); Herman et al. (2016); Kostrikov et al. (2019); Swamy et al. (2021); Florence et al. (2022). However, since these methods imitate all given demonstrations, they often require a large amount of clean expert data, which can be expensive for real-world tasks.

Recently, there has been growing interest in exploring how to effectively leverage imperfect data in offline IL (Xu et al., 2022a; Yu et al., 2022; Sasaki and Yamashina, 2020; Kim et al., 2022). Sasaki and Yamashina (2020) analyze why the imitation policy trained by BC deteriorates its performance when using noisy demonstrations. They reuse an ensemble of policies learned from the previous iteration as the weight of the original BC objective to extract the expert behaviors. However, this requires that expert data occupies the majority proportion of the offline dataset, otherwise the policy will be misguided to imitate the suboptimal data. Kim et al. (2022) retrofit the BC objective with an additional KL-divergence term to regularize the learned policy to stay close to the behavior policy. Although it can implicitly extract the behaviors that bear similarity to the expert demonstrations, it easily fails to achieve satisfactory performance when the diverse data is highly suboptimal. Xu et al. (2022a) cope with this issue by introducing an additional discriminator, the outputs of which serve as the weights of the original BC loss, so as to imitate demonstrations selectively. Unfortunately, it selects behaviors building on state-action similarity, which does not suffice to leverage the dynamics information and diverse behaviors. In offline RL, Yu et al. (2022) propose to utilize unlabeled data by applying zero rewards, but this method necessitates a large amount of labeled offline data. In contrast, this paper focuses on the setting with no access to any reward signals.

Offline inverse reinforcement learning (IRL) explicitly learns a reward function from offline datasets, aiming to comprehend and generalize the underlying intentions behind expert actions (Lee et al., 2019). Zolna et al. (2020) propose ORIL that constructs a reward function that discriminates expert and exploratory trajectories, followed by an offline RL progress. Chan and van der Schaar (2021) use a variational method to jointly learn an approximate posterior distribution over the reward and policy. Garg et al. (2021) propose to learn a soft Q-function that implicitly represents both reward and policy, which can stabilize the training. To cope with the reward extrapolation error, Chang et al. (2022) introduce a model-based offline IRL algorithms that uses a model inaccuracy estimate to penalize the learned reward function on out-of-distribution state-actions. Recently, Yue et al. (2023) also propose a model-based offline IRL approach, named CLARE. In contrast to Chang et al. (2022), they compute a conservative element-wise weight to implicitly penalize out-of-distribution behaviors. However, it is highly challenging to define and learn meaningful reward functions without environmental interaction (Xu et al., 2022b). The model-based approaches often struggle to scale in high-dimensional environments, and their min-max progress usually causes training to be unstable and inefficient.

## 6 Conclusion

In this paper, we introduce a simple yet effective data selection method along with a lightweight behavior cloning algorithm to fully leverage the imperfect demonstrations in offline IL. In contrast to the prior methods, we exploit the resultant states to access the value of behaviors, which is an informative criterion that enables explicit utilization of dynamics information and the extraction of both expert-like and beneficial diverse behaviors. We provide necessary theoretical guarantees for the proposed method, and extensive experiments corroborate that iLID outperforms existing methods in continuous, high-dimensional environments by a significant margin. In future work, we plan to establish theoretical guarantees for iLID in the general stochastic MDPs and explore whether the proposed methods can benefit offline RL in terms of data selection and policy optimization.

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
