# OpenReview forum: "How to Leverage Imperfect Demonstrations in Offline Imitation Learning"
_NeurIPS.cc/2023/Conference — Submitted to NeurIPS 2023_

### Official Review · Reviewer_Y7ma · 2023-06-24

**Soundness:** 3 good
**Presentation:** 2 fair
**Contribution:** 3 good
**Rating:** 7
**Confidence:** 3

**Summary:**

This paper addresses the challenge of learning good imitative policies from offline data, in which abundant imperfect demonstrations are mixed with few expert ones. Unlike previous work that measures the state-action similarity between imperfect and expert data, the present work proposes iLID, which leverages trajectories in imperfect data that lead to expert states in several steps. The sample complexity analysis indicates that this approach benefits the performance of imitation policy, and the empirical results suggest that iLID outperforms baselines including state-of-the-art offline imitation learning methods.

**Strengths:**

* As illustrated in Figure 1, the original idea of selecting imperfect demonstrations leading to expert states is novel and makes intuitive sense.
* The policy optimization problem is well-posed, and is straightforward to implement with alternating dual ascent.
* Empirical results suggest a large performance gain for iLID compared to state-of-the-art baselines, especially when the dataset contains very few expert demonstrations. In particular, the ablation study in Figure 3 does a good job of explaining why the constrained optimization problem leads to a better policy than the naive direct imitation approach.


**Weaknesses:**

The quality of presentation can be improved.
1) $\tilde{\mathcal{D}}$ in equation (6) overloads the notation that was originally presented in Section 3.1 without time indices.
2) Remarks in Section 3.1 state that the sample complexity of the proposed approach is better than the vanilla BC, but there’s no citation for the BC sample complexity.
3) The explanation on the behavior interference for the complementary dataset $\tilde{\mathcal{D}}$ did not make full sense and requires further clarification. In particular, it is unclear why more recent actions are preferred when the same state appears multiple times in the trajectory, even though the underlying MDP does not have any discount factor in the definition of the value function. (What would happen if the discount factor $\gamma$ in equation (7) is set to 1 for all the experiments?)

**Questions:**

* In Figure 4 and Table 1, it appears that all the methods resulted in poor performance for the Door task. Is it a failure case, and if so what was so difficult about the task?
* It was not easy for me to follow the proof of Theorem 3.1. There are quite a few questions about the proof steps:
1) What is the outer expectation over in the definition of $\epsilon$ and $\delta$? Is it over the randomness in the choice of the datasets? If so, do you require any assumptions on the distributions of $\mathcal{D}_e$ and $\tilde{\mathcal{D}}$, such as independence?
2) How do you get from the 1st inequality to the 2nd in equation (13)? Specifically, how do you show $\mathbb{E}\left[\mathbb{1}(s \notin \mathcal{S}_1(\mathcal{D}_e)) \mathbb{1}(s \notin \mathcal{S}_1(\tilde{\mathcal{D}}))\right] \leq \mathbb{E}\left[\mathbb{1}(s \notin \mathcal{S}_1(\mathcal{D}_e))\right] \mathbb{E}\left[\mathbb{1}(s \notin \mathcal{S}_1(\tilde{\mathcal{D}}))\right]$?
3) Whether $\epsilon = \mathbb{E}\left[\mathbb{1}(s \notin \mathcal{S}_1(\mathcal{D}_e))\right]$ and $\delta = \mathbb{E}\left[\mathbb{1}(s \notin \mathcal{S}_1(\tilde{\mathcal{D}}))\right]$ hold or not seem to depend on the distribution of $ \mathcal{D}_e $ and $ \mathcal{\tilde{D}} $. For example, $\epsilon = \mathbb{E}\left[ \mathbb{E}_s [ \mathbb{1}(s \notin \mathcal{S}_1(\mathcal{D}_e)) ]\right] = \mathbb{E}_s \left[ \mathbb{E} [ \mathbb{1}(s \notin \mathcal{S}_1(\mathcal{D}_e)) ]\right] = \sum_s \frac{1}{|S|} \mathbb{E} [\mathbb{1}(s \notin \mathcal{S}_1(\mathcal{D}_e)) ] = \sum_s \frac{1}{|S|} \mathbb{P}(s \notin \mathcal{S}_1(\mathcal{D}_e))$ , and the quantity $\mathbb{P}(s \notin \mathcal{S}_1(\mathcal{D}_e))$ may differ for different $s \in \mathcal{S}$ depending on the distribution of $\mathcal{D}_e$.
4) The 1st equality of equation (15) is not obvious. How do you show that $\mathbb{E}\left[\mathbb{1}(s \notin \mathcal{S}_1(\mathcal{D}_e)) \mathbb{1}(s \in \mathcal{S}_1(\tilde{\mathcal{D}}))\right] = \mathbb{E}\left[\mathbb{1}(s \notin \mathcal{S}_1(\mathcal{D}_e))\right] \mathbb{E}\left[\mathbb{1}(s \in \mathcal{S}_1(\tilde{\mathcal{D}}))\right]$? Do you assume that the events $\mathbb{1}(s \notin \mathcal{S}_1(\mathcal{D}_e))$ and $\mathbb{1}(s \in \mathcal{S}_1(\tilde{\mathcal{D}}))$ are independent for all $s \in \mathcal{S}$?
4) The 1st inequality of equation (15) does not make sense. Did you mean $\epsilon(1 - \delta) V^{\pi_e}$ rather than $\epsilon(1 - \delta) V^{\pi_e}(s)$? Where does the inequality come from?
5) Can you elaborate on how to derive $\mathbb{E}_{s' \sim \tilde{D}(\cdot | s)} V'(s')$ in equation (16)? Does it hold because we only have to consider the case $s \in \mathcal{S}_1(\mathcal{\tilde{D}}) \backslash \mathcal{S}_1(\mathcal{D}_e)$ there, which would correspond to the 2nd case in equation (2)?

Please elaborate on the points above and clarify any underlying assumptions that are used implicitly.

**Limitations:**

The authors have not explicitly provided limitations in the paper. One conceivable limitation is that one requires the datasets of expert and imperfect demonstrations to be labeled as such, although it seems unavoidable for any methods of this kind. Another limitation could be that the resulting policy may still fail to discover diverse modes to accomplish the task, if the expert demonstration only has a single mode. For instance, in a goal-reaching navigation task similar to the one depicted in Figure 1, the expert policy still needs to exhibit the two different paths so that iLID learns to discover both modes to approach the goal.

---

> ### Author Rebuttal · Authors · 2023-08-09
>
> Thank you for your appreciation of the contribution of this paper! Below are detailed responses to each comment:
>
> ---
> ## Q1: About the proof steps
>
> **(1) What is the outer expectation over in the definition of $\epsilon$ and $\delta$? Do you require any assumptions on the distributions of $\mathcal{D}_e$ and $\tilde{D}$**
>
> Regarding the first question, the reviewer's understanding is accurate: the outer expectation is taken w.r.t the randomness in the choice of datasets $\mathcal{D}_e$ and $\tilde{D}$. Regarding the second question, as stated in Section 2, the tuples in $\mathcal{D}_e$ are assumed to be sampled from the expert's state-action distribution, and for tuple $(s,a,s')\in\tilde{D}$, $s$ and $s'$ follow $\mu$ and $\mathcal{S}(\mathcal{D}_e)$ respectively. Accordingly, the *starting* states in the state-action tuples of $\tilde{D}$ are independent of $\mathcal{D}_e$, while the resultant states are contingent on $\mathcal{D}_e$. This assumption empowers us to characterize the diverse transitions leading to given expert states with necessary statistical properties.
>
> **(2) How do you get from the 1st inequality to the 2nd in equation (13)?**
>
> The derivation is from the independence of the distributions of the starting states $\mathcal{S}_1(\mathcal{D}_e)$ and $\mathcal{S}_1(\tilde{\mathcal{D}})$, both of which are independently sampled from the initial state distribution $\mu$ (as elaborated in the preceding question).
>
> **(3) Whether $\mathbb{E}[\mathbb{1}(s\notin\mathcal{S}_1(\mathcal{D}_e))]=\epsilon$ or $\mathbb{E}[\mathbb{1}(s\notin\mathcal{S}_1(\tilde{\mathcal{D}}))]=\delta$ hold or not seem to depend on the distributions of $\mathcal{D}_e$ and $\tilde{\mathcal{D}}$.**
>
> The reviewer's understanding is correct, and it is indeed the reason why we impose the assumption prior to Theorem 1: the starting states $\mathcal{S}_1(\mathcal{D}_e)$ and $\mathcal{S}_1(\tilde{\mathcal{D}})$ follow the uniform distribution $\mu$. Notably, this assumption can be relaxed to yield a more generalized result, as detailed in our response to Reviewer 71T4.
>
> **(4) The 1st inequality of equation (15) does not make sense.**
>
> Sorry for the typo. The correct expression in equation (15) should be $\epsilon (1-\delta)V^{\pi_e}$. We will diligently review the proof and rectify any such errors.
>
> **(5) Elaboration on how to derive $\mathbb{E}_{s'\sim\tilde{\mathcal{D}}(\cdot|s)}V'(s')$ in equation (16).**
>
> Denote $\mathcal{D}_c\doteq\mathcal{S}_1(\tilde{\mathcal{D}})/\mathcal{S}_1(\mathcal{D}_e)$. We detail the derivation from the 3rd equality to the 4th equality as follows:
>
> $$\mathbb{E}\_{s\sim\mu} \Big[\mathbb{1}(s\in\mathcal{D}_c) \mathbb{E}\_{a\sim\tilde{\pi}(\cdot|s),s'\sim T(s,a)}\big[V'(s')\big]\Big]$$
> $$=\sum\_{s\in\mathcal{D}_c}\mu(s) \mathbb{E}\_{a\sim\tilde{\pi}(\cdot|s),s'\sim T(s,a)}\big[V'(s')\big]$$
> $$=\sum\_{s\in\mathcal{D}_c}\mu(s)\sum\_a \frac{\sum\_{(\tilde{s},\tilde{a},\tilde{s}')\in\tilde{\mathcal{D}}}\mathbb{1}((\tilde{s},\tilde{a})=(s,a))}{\sum\_{(\tilde{s},\tilde{a},\tilde{s}')\in\tilde{\mathcal{D}}}\mathbb{1}(\tilde{s} = s)} V'(T(s,a))$$
> $$=\sum\_{s\in\mathcal{D}_c}\mu(s)\frac{\sum\_{(\tilde s,\tilde a, \tilde s')\in\tilde{\mathcal{D}}}V'(\tilde s')\sum\_a\mathbb{1}((\tilde s,\tilde a)=(s,a))}{\sum\_{(\tilde{s},\tilde{a},\tilde{s}')\in\tilde{\mathcal{D}}}\mathbb{1}(\tilde{s} = s)}$$
> $$=\sum\_{s\in\mathcal{D}_c}\mu(s)\frac{\sum\_{(\tilde s,\tilde a, \tilde s')\in\tilde{\mathcal{D}}}\mathbb{1}(\tilde s=s)V'(\tilde s')}{\sum\_{(\tilde{s},\tilde{a},\tilde{s}')\in\tilde{\mathcal{D}}}\mathbb{1}(\tilde{s} = s)}$$
> $$=\sum\_{s\in\mathcal{D}_c}\mu(s)\frac{\sum\_{(\tilde s,\tilde a, \tilde s')\in\tilde{\mathcal{D}}}\mathbb{1}(\tilde s=s)\sum\_{s'}\mathbb{1}(\tilde s'=s')V'(s')}{\sum\_{(\tilde{s},\tilde{a},\tilde{s}')\in\tilde{\mathcal{D}}}\mathbb{1}(\tilde{s} = s)}$$
> $$=\sum\_{s\in\mathcal{D}\_c}\mu(s)\sum\_{s'}\frac{\sum\_{(\tilde s,\tilde a, \tilde s')\in\tilde{\mathcal{D}}}\mathbb{1}(\tilde s=s,\tilde s'=s')V'(s')}{\sum\_{(\tilde{s},\tilde{a},\tilde{s}')\in\tilde{\mathcal{D}}}\mathbb{1}(\tilde{s}=s)}$$
> $$=\sum\_{s\in\mathcal{D}_c}\mu(s)\mathbb{E}\_{s'\sim\tilde{\mathcal{D}}(\cdot|s)}\big[V'(s')\big]$$
> $$=\mathbb{E}\_{s\sim\mu}\Big[\mathbb{1}(s\in\mathcal{D}_c)\mathbb{E}\_{s'\sim\tilde{\mathcal{D}}(\cdot|s)}\big[V'(s')\big]\Big].$$
>
> ---
> ## Q2: About presentation
>
> **(1) $\tilde{\mathcal{D}}$ in equation (6) overloads the notation that was originally presented in Section 3.1 without time indices.**
>
> To clarify it, we have added an explanation, "Slightly abusing notation, we denote $\tilde{\mathcal{D}}$ as the buffer of selected diverse data", in the data selection section. We will undertake a check of notations used throughout the paper and provide the appropriate explanation to ensure clarity and consistency.
>
> **(2) Lack of citation for the sample complexity of BC.**
>
> The result can be found in Theorem 2 of (Xu et al. 2021). We will revise the manuscript accordingly.
>
> (Xu et al. 2021) [On generalization of adversarial imitation learning and beyond]([arxiv.org/pdf/2106.10424.pdf](https://arxiv.org/pdf/2106.10424.pdf)).
>
> **(3) Why are more recent actions preferred when the same state appears multiple times in the trajectory?**
>
> To clarify, we add ablation studies on all 4 MuJoCo environments by setting $\gamma=1$. As implied in Figure 3 in the PDF, without properly controlling the priorities among selected behaviors, the actions necessitating multiple steps to reach expert states may suffer from vulnerability to environmental ***uncertainty*** and hamper performance.
>
> ---
> ## Q3: About the Door task
>
> The door task involves undoing the latch and swinging the door open. The latch has significant dry friction and a bias torque that forces
> the door to stay closed. Without environmental interaction, it is highly challenging for the offline agent to develop an understanding of the latch as no information about the latch is explicitly provided, and the position of the door is also randomized.
>
> ---

---

> ### Comment · Reviewer_Y7ma · 2023-08-20
>
> Dear authors,
>
> Thank you for the detailed clarification, and I apologize for the delay in my response. The math in the proof of Theorem 3.1 is much clearer now. I was originally concerned that the uniform distribution assumptions on $\mathcal{S}_1(D_e)$ and $\mathcal{S}_1(\tilde{D})$ would be too strong. However, it is nice that the authors have been able to relax the assumption to yield a similar (albeit looser) bound in the response to reviewer 71T4.
>
> >To clarify, we add ablation studies on all 4 MuJoCo environments by setting $\gamma = 1$. As implied in Figure 3 in the PDF, without properly controlling the priorities among selected behaviors, the actions necessitating multiple steps to reach expert states may suffer from vulnerability to environmental uncertainty and hamper performance.
>
> Aside from the typo in Figure 3 in the new PDF (you meant $\gamma$, not $\lambda$, didn't you?), it visually describes, to some extent, that more recent causal actions leading to expert states are preferred when *stochasticity* exists in the environment . My original confusion was caused by the discrepancy between the theoretical part of the paper assuming deterministic transitions and the algorithmic part assuming a stochastic environment. I encourage the authors to make the distinction clear in writing the final version of the manuscript. Nevertheless, the authors have addressed most of the concerns I have had in the original review, and thus I will leave the score unchanged. The only minor comment I have is on the limitations of the work. I am curious whether the authors agree with the potential limitations that I wrote in my initial review, and if so I would appreciate it if they mentioned them in the final version of the paper.

---

> > ### Author Response · Authors · 2023-08-21
> >
> > We deeply appreciate the reviewer's meticulous reexamination of our responses and the in-depth suggestions for refining our work. Regarding Figure 3 in the attached PDF, we are sorry for the typo, and the correct notation should be $\gamma$ instead of $\lambda$. Regarding the alignment between the theoretical motivation and the algorithmic design, we will emphasize this point in the revised version. As an example, we have augmented Section 3.2 with an elucidation, "it is worth noting that while our theoretical motivation is framed in deterministic cases, iLID can be applied to general stochastic environments." Regarding the limitations, we concur with the reviewer's remarks and will integrate them into our final version. In particular, the second mentioned limitation actually stems from the isolation between the manifolds of expert and suboptimal data. With no state similarity between expert data and suboptimal data, our algorithm can hardly abstract positive multi-mode diverse behaviors, and its performance will reduce toward BCE. In contrast, if a different mode for achieving the goal exists in the suboptimal data, even with a single-mode expert demonstration, our approach can effectively identify this alternative mode from the causal behaviors of the goal state (overlapping with the expert states). More specifically, we will add the following limitation section to the revised manuscript.
> >
> > ## Limitation of iLID
> >
> > The main limitation of this paper lies in the requirement of the existence of state similarity between the suboptimal data and the expert data. Without any state similarity, our algorithm can hardly abstract positive or multi-mode diverse behaviors, and the performance will reduce to BC. However, it is worth noting that in the context of offline IL, if the states within suboptimal data deviate significantly from the given expert data, it is extremely challenging to assess the values of suboptimal behaviors with no prior information, because the non-expert behaviors can deviate arbitrarily from desirable behaviors.
> >
> > Another limitation of this work is that one requires the datasets of expert and imperfect demonstrations to be labeled. Our method may not reason about the stochasticity of the expert behaviors. For example, humans can take suboptimal behaviors when these behaviors bear lesser importance in the task (Ziebart et al. 2008). In addition, there is a lack of theoretical guarantees in the general MDPs, which we will continue to explore by utilizing the analytical idea introduced in this work and replacing the initial distribution with the state-action occupancy measure.
> >
> > **Reference**
> >
> > (Ziebart et al. 2008) [Maximum Entropy Inverse Reinforcement Learning](https://cdn.aaai.org/AAAI/2008/AAAI08-227.pdf)

---

### Official Review · Reviewer_71T4 · 2023-06-25

**Soundness:** 1 poor
**Presentation:** 3 good
**Contribution:** 2 fair
**Rating:** 3
**Confidence:** 4

**Summary:**

The submission presents a novel method called Offline Imitation Learning with Imperfect Demonstrations (iLID) for Offline Imitation Learning, which aims to improve policy learning from both expert and imperfect demonstrations. Compared with previous IL methods, which only consider the state-action pairs during learning, this paper also considers the dynamics of the non-expert data. To this end, the submission proposed employs a data selection technique using a discriminator on the resultant state of behavior, meanwhile integrating lightweight-constrained behavior cloning. Some empirical studies showed the outperformance of the proposed method compared with other baselines.

**Strengths:**

1. Overall the paper is well-written.
2. The motivation for including dynamics in behavior cloning makes sense and easy-to-follow.
3. The experimental tasks and baselines are sufficient.

**Weaknesses:**

1. Some notations are so confusing that the descriptions of these notations can not lead to corresponding theoretical results.
2. The assumption of Theorem 3.1 is too strong but without any discussion.
3. The proposed algorithm and the motivation in the introduction (Figure 1) are isolated.

**Questions:**

1. In line 76 the initial state distribution $\mu$ is defined as the mapping function $\mathcal{S} \rightarrow [0, 1]$. Then how is it possible to sample $s_i \sim \mu$ but $s'_i \sim \mathcal{S}(\mathcal{D}_e)$ in the definition of $\widetilde{\mathcal{D}}$? After that, in Theorem 3.1, $\mu$ is again defined by $U(\mathcal{S})$, but its meaning is completely different from the meaning of the initially defined mapping function.
2. $\mu$ is usually defined as a subset of the state space $\mathcal{S}$. The assumption in Theorem 3.1 that $\mu=U(\mathcal{S})$ is too strong and almost impossible for decision-making environments.
3. There exists no empirical study or statement to illustrate how the proposed algorithm reflects the motivation in the introduction.

**Limitations:**

There is no discussion about the limitation or societal impact in the submission.

---

> ### Author Rebuttal · Authors · 2023-08-09
>
> Thank you for your valuable and detailed feedback! Below are detailed responses to each comment, and new comments on them are very welcome!
>
> ---
>
> ## Q1: About the notation, $\mu$
>
> We are sorry for this typo in the preliminary section. Throughout the paper, $\mu$ is used as a distribution or overloaded as the corresponding probability measure (function). We will diligently review the manuscript and rectify any such errors.
>
> ---
> ## Q2: About the assumption, $\mu=U(\mathcal{S})$
>
> Thank you for pointing this out. It is important to highlight that we introduce this assumption primarily for the characterization of achieving strong performance at each individual state, especially within the context of this deterministic motivational setting. As suggested by the reviewer, we proceed to remove this assumption and derive more generalized results as follows.
>
> Note that the assumption is only used in Eqs. (13), (15), and (16) in the Appendix. Next, we refine Eqs. (13), (15), and (16) by removing the assumption. Denote the maximum probability of the initial state not being in $\mathcal{S}_1(\mathcal{D}_e))$ and the minimum probability of the initial state not being in $\mathcal{S}_1(\tilde{\mathcal{D}}))$ as
>
> $$\epsilon_{\mathrm{max}}\doteq \max \max \Big\\{\mathbb{E}\_{\mathcal{D}_e}\big[\mathbb{1}(s\notin\mathcal{S}_1(\mathcal{D}_e))\big]\,\Big\vert\,\mu(s)>0,s\in\mathcal{S}\Big\\},\quad \delta\_{\mathrm{min}}\doteq\min \Big\\{\mathbb{E}\_{\tilde{\mathcal{D}}}\big[\mathbb{1}(s\notin\mathcal{S}_1(\tilde{\mathcal{D}}))\big] \,\Big\vert\, \mu(s)>0,s\in\mathcal{S}\Big\\}.$$
>
> Regarding Eq. (13), we have
>
> $$\mathbb{E}\left[(a)\right]$$
> $$=\mathbb{E}\Big[\mathbb{E}\_{s\sim\mu}\Big[\mathbb{1}(s\notin\mathcal{S}_1(\mathcal{D}_e))\cdot\mathbb{1}(s\notin\mathcal{S}\_1(\tilde{\mathcal{D}}))\cdot\big( V^{\pi_e}(s) - V^{\tilde{\pi}}(s)\big)\Big]\Big]$$
> $$\overset{(a)}{\le} H\mathbb{E}\left[\mathbb{E}\_{s\sim\mu}\left[\mathbb{1}(s\notin\mathcal{S}_1(\mathcal{D}_e))\cdot\mathbb{1}(s\notin\mathcal{S}_1(\tilde{\mathcal{D}}))\right]\right]$$
> $$\overset{(b)}{\le} H\mathbb{E}\_{s\sim\mu}\left[\mathbb{E}\big[\mathbb{1}(s\notin\mathcal{S}_1(\mathcal{D}_e))\big]\cdot\mathbb{E}\big[\mathbb{1}(s\notin\mathcal{S}_1(\tilde{\mathcal{D}}))\big]\right]$$
> $$\overset{(c)}{\le} H \sqrt{\mathbb{E}\big[\mathbb{E}\big[\mathbb{1}(s\notin\mathcal{S}_1(\mathcal{D}_e))\big]^2\big] \cdot\mathbb{E}\big[\mathbb{E}\big[\mathbb{1}(s\notin\mathcal{S}_1(\tilde{\mathcal{D}}))\big]^2\big]}$$
> $$\overset{(d)}{\le} H \sqrt{\mathbb{E}\big[\mathbb{E}\big[\mathbb{1}(s\notin\mathcal{S}_1(\mathcal{D}_e))^2 \big]\big]\cdot\mathbb{E}\big[\mathbb{E}\big[\mathbb{1}(s\notin\mathcal{S}_1(\tilde{\mathcal{D}}))^2 \big]\big]}$$
> $$= H \sqrt{\mathbb{E}\big[\mathbb{E}\big[\mathbb{1}(s\notin\mathcal{S}_1(\mathcal{D}_e))\big]\big]\cdot\mathbb{E}\big[\mathbb{E}\big[\mathbb{1}(s\notin\mathcal{S}_1(\tilde{\mathcal{D}})) \big]\big]}$$
> $$=H\sqrt{\delta\epsilon},$$
> where $(a)$ is from $V^\pi(s)\le H$, $(b)$ from the independency of  $\mathcal{S}_1(\mathcal{D}_e))$ and $\mathcal{S}_1(\tilde{\mathcal{D}}))$, $(c)$ from the Cauchy-schwarz inequality, and $(d)$ from $\mathbb{E}[X]^2\le\mathbb{E}[X^2]$. Regarding Eq. (15), the following holds:
> $$\mathbb{E}\Big[\mathbb{E}\_{s\sim\mu}\Big[\mathbb{1}(s\notin\mathcal{S}_1(\mathcal{D}_e))\cdot\mathbb{1}(s\in\mathcal{S}_1(\tilde{\mathcal{D}}))\cdot V^{\pi_e}(s)\Big]\Big]$$
> $$=\mathbb{E}\_{s\sim\mu}\Big[\mathbb{E} \big[\mathbb{1}(s\notin\mathcal{S}\_1(\mathcal{D}\_e))\big]\cdot\mathbb{E}\big[\big(1-\mathbb{1}(s\notin\mathcal{S}\_1(\tilde{\mathcal{D}}))\big)\big]\cdot V^{\pi\_e}(s)\Big]$$
> $$\le \epsilon\_\max (1-\delta\_\min)V^{\pi_e},$$
> where the last inequality is from the definitions of $\epsilon\_\max$ and $\delta\_\min$. Similarly, for Eq. (16), it can be easily seen that
> $$\mathbb{E}\left[\mathbb{E}\_{s\sim\mu}\left[\mathbb{1}(s\notin\mathcal{S}_1(\mathcal{D}\_e))\cdot\mathbb{1}(s\in\mathcal{S}\_1(\tilde{\mathcal{D}}))\cdot  V^{\tilde{\pi}}(s)\right]\right]\le\epsilon\_\max (1-\delta\_\min)\mathbb{E}\_{s'\sim\rho^{\pi\_e}}\big[V'(s')\big].$$
>
> Using the above results and following the proof steps in Theorem 1, we can obtain a generalized result independent with $\mu=U(\mathcal{S})$ as:
>
> $$V^{\pi\_e} - \mathbb{E}\big[V^{\tilde{\pi}}\big] \le \left(\frac{H}{2}+\frac{1}{H}\right)(1-\delta\_\min)\epsilon\_\max + H\sqrt{\delta\epsilon}$$
>
> where $\epsilon$ and $\delta$ are defined in Theorem 1. As $\delta\_\min$ and $\delta$ rely on the data coverage of $\tilde{\mathcal{D}}$, this aligns with the fact that a large $\tilde{\mathcal{D}}$ can effectively combat the error accumulation in BC.
>
> ---
> ## Q3: About motivations
>
> We respectfully disagree with the reviewer about "there exists no empirical study or statement to illustrate how the proposed algorithm reflects the motivation in the introduction." Figure 1 showcases the benefit (generalization to unseen states) of ***the diverse behaviors that can lead to expert states***. Section 3 is dedicated to illustrating our methodology, which learns ***a state-only identifier to distinguish positive diverse behaviors based on their resultant states*** and leverage the selected data properly. Our empirical evaluation is carried out in a setting where the agent can only get access to notably limited expert data (1 trajectory) along with (low-quality) mixed suboptimal data. The pronounced performance superiority of our method can serve as compelling evidence, underscoring the algorithm's efficacy in extracting positive behaviors from suboptimal data and substantiating the alignment between our proposed approach and the motivations outlined in the introduction.
>
> ---

---

> > ### Comment · Reviewer_71T4 · 2023-08-17
> >
> > I appreciate the authors' responses and further discussions. However, I still have some concerns regarding the second and third questions.
> >
> > ## Q2: The Updated Theorem 3.1
> >
> > After the removal of the strong assumption that $\mu \in U(\mathcal{S})$, the updated expectation term in (a) is bounded by $H\sqrt{\delta\epsilon}$, which is even tighter than the original bound in (a), i.e., $H\delta\epsilon$. I am confused about how this is possible. Also, to obtain the last equation in (d), don't we still need the assumption that $\mu \in U(\mathcal{S})$?
> >
> > ## Q3: Motivation and Methodology
> >
> > The motivation of this work, as presented in Figure 1, is that "when the agent encounters a state unobserved in expert demonstrations, compared to taking a random action, a more reasonable way is to return to the states where it knows expert behaviors." My concern is that there is no empirical study to support this statement. In the experiments (including the appendix), there are only performance comparisons. Does the non-expert data singled out by the proposed method actually help guide the policy towards expert data? What if there is no overlap between the non-expert and expert data?

---

> > > ### Author Response · Authors · 2023-08-18
> > >
> > > We appreciate the reviewer's further response and in-depth clarification. Below are detailed responses to each follow-up comment, and new comments on them are very welcome!
> > >
> > > ---
> > > ## Q2: The Updated Theorem 3.1
> > >
> > > **(1) Is the updated bound of (a) tighter than the original bound?**
> > >
> > > The answer is no. Please recall the definitions $\epsilon\doteq\mathbb{E}[\mathbb{E}\_{s\sim\mu}[\mathbb{1}(s\notin\mathcal{S}_1(\mathcal{D}_e))]]$ and $\delta\doteq\mathbb{E}[\mathbb{E}\_{s\sim\mu}[\mathbb{1}(s\notin\mathcal{S}_1(\tilde{\mathcal{D}}))]]$, which indicates $\epsilon,\delta\in[0,1]$. The updated bound $H\sqrt{\delta\epsilon}$ is in fact looser than the original bound $H\delta\epsilon$.
> > >
> > > **(2) Do we still need $\nu=U(\mathcal{S})$ to obtain the last equation in (d)?**
> > >
> > > The answer is no. To clarify, we elaborate on the derivation from (c) to (d) below. First, the inequality (d) is derived from the fact $\mathbb{E}[X]^2\le\mathbb{E}[X^2]$:
> > > $$\mathbb{E}[(a)]\overset{(c)}{\le}H\sqrt{\mathbb{E}\_{s\sim\mu}\big[\mathbb{E}\big[\mathbb{1}(s\notin\mathcal{S}_1(\mathcal{D}_e))\big]^2\big]\cdot\mathbb{E}\_{s\sim\mu}\big[\mathbb{E}\big[\mathbb{1}(s\notin\mathcal{S}_1(\tilde{\mathcal{D}}))\big]^2\big]}\overset{(d)}{\le} H \sqrt{\mathbb{E}\_{s\sim\mu}\big[\mathbb{E}\big[\mathbb{1}(s\notin\mathcal{S}_1(\mathcal{D}_e))^2\big]\big]\cdot\mathbb{E}\_{s\sim\mu}\big[\mathbb{E}\big[\mathbb{1}(s\notin\mathcal{S}_1(\tilde{\mathcal{D}}))^2\big]\big]}.$$
> > >
> > > Then, using the facts $\mathbb{1}(s\notin\mathcal{S}_1(\mathcal{D}_e))^2 =\mathbb{1}(s\notin\mathcal{S}_1(\mathcal{D}_e))$ and $\mathbb{1}(s\notin\mathcal{S}_1(\tilde{\mathcal{D}}))^2=\mathbb{1}(s\notin\mathcal{S}_1(\tilde{\mathcal{D}}))$, we have
> > > $$H\sqrt{\mathbb{E}\_{s\sim\mu}\big[\mathbb{E}\big[\mathbb{1}(s\notin\mathcal{S}_1(\mathcal{D}_e))^2 \big]\big]\cdot\mathbb{E}\_{s\sim\mu}\big[\mathbb{E}\big[\mathbb{1}(s\notin\mathcal{S}_1(\tilde{\mathcal{D}}))^2 \big]\big]}=H \sqrt{\mathbb{E}\_{s\sim\mu}\big[\mathbb{E}\big[\mathbb{1}(s\notin\mathcal{S}_1(\mathcal{D}_e)) \big]\big]\cdot\mathbb{E}\_{s\sim\mu}\big[\mathbb{E}\big[\mathbb{1}(s\notin\mathcal{S}_1(\tilde{\mathcal{D}})) \big]\big]}=H\sqrt{\delta\epsilon},$$
> > >
> > > where the second equality holds from the definitions of $\epsilon$ and $\delta$. The assumption $\mu=U(\mathcal{S})$ is never used in the above proof.
> > >
> > > ---
> > > ## Q3: Motivation and Methodology
> > >
> > > **(1) Does the non-expert data singled out by the proposed method actually help guide the policy toward expert data?**
> > >
> > > Since our proposed method selects ***the causal behaviors*** of identified expert states, the extracted non-expert training data are indeed the ***sub-trajectories that lead to expert states***. Therefore, it is natural that the data (toward expert states) will guide the learned policy toward expert states in non-expert environments.
> > >
> > > To better illustrate the efficacy of our method in guiding the policy toward expert states, we further examine ***the ratio of expert states*** in the policies' visited states of BC and our method. The experiments are carried out under the settings of Table 1, and we regard the states with the discriminator output $d(s)>0.5$ as the (approximated) expert states. The results below correspond to the ratio of visited expert states across ten episodes.
> > >
> > > | Environment | BC | iLID (ours) |
> > > | :---------- | :----- | :---------- |
> > > | Ant         | 7.98%  | 57.83%      |
> > > | HalfCheetah | 3.67%  | 25.07%      |
> > > | Hopper      | 5.91%  | 78.78%      |
> > > | Walker2d    | 53.14% | 90.52%      |
> > >
> > > Due to learning only on given expert data, BC would probably take random actions in the states beyond the given expert data manifold. The results demonstrate that our method enjoys a higher incidence of visits to expert states, which can affirm the claim.
> > >
> > > **(2) What if there is no overlap between the non-expert and expert data?**
> > >
> > > If there is no similarity between the non-expert and expert ***states***, without any prior knowledge of the non-expert data, the performance of our algorithm would reduce to BC. It corresponds to the complementary experiment on Halfcheetah with the Random data and a single expert trajectory, where our method only selects 100 non-expert state-actions with just one identified expert state. However, it is important to note that assessing values of suboptimal behaviors in this case is extremely challenging for offline IL, because the non-expert behaviors can deviate arbitrarily from desirable behaviors (as illustrated in Figure 1 in the attached PDF, all baselines fail in this case).
> > >
> > > In addition, the existence of state similarity (even state-action similarity) between the non-expert and expert data is a commonly used assumption in existing works of offline IL ([Kim et al. 2022](https://openreview.net/pdf?id=BrPdX1bDZkQ);[Xu et al. 2022](https://proceedings.mlr.press/v162/xu22l/xu22l.pdf)). As shown in the complementary results, it is also applicable in widely used RL benchmarks (please refer to our response to Reviewers b1iR and K4fk for more details).

---

### Official Review · Reviewer_b1iR · 2023-07-05

**Soundness:** 2 fair
**Presentation:** 4 excellent
**Contribution:** 3 good
**Rating:** 4
**Confidence:** 3

**Summary:**

The paper addresses the problem of offline imitation learning (IL) from demonstrations that are noisy/suboptimal. To this end, the authors propose iLID which is a two-step process—a data selection step which only retains those $(s,a)$ transitions from suboptimal demonstrations which lead to states in the expert demonstrations, thereby maintaining a supplementary data buffer; then policy learning is performed by behavior cloning on samples from the supplementary buffer while also regularizing the policy to not stray too far from a BC expert policy (on samples from the expert data buffer.)

The authors establish competitive upper bounds on the suboptimality and sample complexity of iLID. Extensive experimental on 8 complex robotic tasks, and accompanying sensitivity studies show that iLID outperforms 5 competing baselines, and limited sensitivity analysis is performed.


**Strengths:**

-	The paper tackles the important but challenging issue of offline imitation learning from suboptimal demonstrations. In this regard, the paper addresses a pertinent problem in the research area.
-	The rationale behind the formulation is simple yet powerful. Specifically, the data selection step trains a discriminator to select only those transitions in the suboptimal data $(s,a) \in \mathcal{D}_\mathcal{s}$ which lead to a state in the expert data $(s,a) \in \mathcal{D}_\mathcal{e}$ within some specified $K$ timesteps. This is a simple way of leveraging offline data to distil only useful knowledge from suboptimal data for policy learning and may aid the agent in correcting towards expert behavior from non-expert states. The formulation of the policy learning step as a regularized version of BC yields demonstrable improvements in training time.
-	Empirical results on the D4RL robotics benchmark dataset are impressive (Table 1) and hold across all but one environment.
-	Overall, the paper is very well-written and provides helpful illustrations and examples to present ideas.

**Weaknesses:**

Experimentally, seeding imperfect dataset with expert data (~1-20%) seems like a strong assumption. Given that the data selection method explicitly selects $(s,a)$ pairs based on whether they lead to expert states $s \in \mathcal{D}_\mathcal{e}$. If the expert and suboptimal trajectories only share the seeded expert transitions i.e., $\mathcal{D}_\mathcal{e} \cap\mathcal{D}_\mathcal{s} = \mathcal{D}_\mathcal{\text{seeded}}$ (a realistic assumption in real-world cases), then the proposed selection criterion will select only the (seeded) “expert” transitions from $\mathcal{D}_\mathcal{s}$ to add to supplementary data $\tilde{\mathcal{D}}$ (since only states from the seeded expert trajectories in $\mathcal{D}_\mathcal{s}$ would lead to successor states in $\mathcal{D}_\mathcal{e}$). In this case does iLID reduce to just pure BC (Pomerlau, 1998) on just expert data with additional policy regularization? Further, Figure 3a also shows that iLID does rely heavily on expert demonstrations for it to perform well. Some ways to address this –
- Experiment showing results when the imperfect data is not seeded with expert demonstrations would best clarify this issue.
- Experiment showing results for varying # of expert trajectories (as in Fig 3a) for different environments (including the unseeded case).
- As an alternative methodology, to bypass seeding, the imperfect demonstration set could be generated rolling out trajectories from the noise-injected expert BC policy $\tilde{\pi}_{\mathcal{e}}$.



**Questions:**

-	The only major concern is seeding suboptimal data with expert data. How does iLID perform across different environments when this is not done at all?
-	(Table 1) Why does BC with expert data (BCE) perform worse than BC with union data (BCU)? Do they use different number of demonstrations/samples? Compounding errors should have affected both?
-	Can the runtime for DemoDICE be included in Figure 5 (missing figure caption?)

**Limitations:**

While the iLID formulation is interesting, some weaknesses associated with data seeding could be discussed in more detail as per suggestions above.

---

> ### Author Rebuttal · Authors · 2023-08-09
>
> Thank you for your in-depth comments and suggestions! Below are detailed responses to each comment, and new comments on them are very welcome!
>
> **Q1: The only major concern is seeding suboptimal data with expert data. How does iLID perform across different environments when this is not done at all?**
>
> Following the reviewer's suggestion, we exclude expert data from the diverse dataset and conduct a series of extensive experiments across varying environments (Ant, Halfcheetah, Hopper, Walker2d), data qualities (Random, Replay, and Medium), and levels of expert data (1, 3, and 5). We show all the results in Figure 1 in the PDF and select two sets of the results in the tables below (above: in Ant, below: in Halfcheetah) Clearly, the results demonstrate that iLID can effectively unearth valuable behaviors even in the absence of expert data within diverse datasets. Furthermore, we would like to underscore a critical point: to enhance the algorithm's exploitation of diverse data, it is paramount to meticulously calibrate the $\sigma$ and $\gamma$ parameter values. When expert data is relatively scarce, it proves beneficial to increase $\sigma$, thus providing more data support during offline learning. On the contrary, when dealing with a relatively large number of expert data, a reduction in $\sigma$ is more judicious. Moreover, it's worth noting that lower-quality diverse data often correlates with diminished $\gamma$, which aids in mitigating the challenges of behavior interference.
>
> | Suboptimal traj. | # expert traj. |  BCE  |  BCU  | DWBC  | CLARE | DemoDICE  | iLID   (ours) |
> | :--------------: | :------------: | :---: | :---: | :---: | :---: | :-------: | :-----------: |
> |      Random      |       1        | -2.37 | 31.55 | 2.86  | 31.73 |   32.03   |   **33.60**   |
> |      Random      |       3        | -2.18 | 31.48 | 23.67 | 17.71 | **49.63** |     45.13     |
> |      Random      |       5        | 32.00 | 31.48 | 41.46 | 31.75 |   39.48   |   **64.97**   |
> |      Replay      |       1        | -2.37 | 67.73 | 9.76  | 61.42 |   72.74   |   **80.20**   |
> |      Replay      |       3        | -2.18 | 69.82 | 30.11 | 64.33 |   79.97   |   **96.00**   |
> |      Replay      |       5        | 32.00 | 62.93 | 24.61 | 62.72 |   82.17   |   **97.65**   |
> |      Medium      |       1        | -2.37 | 87.67 | -0.08 | 86.24 | **92.07** |     82.96     |
> |      Medium      |       3        | -2.18 | 81.89 | 8.95  | 85.78 |   81.70   |   **89.33**   |
> |      Medium      |       5        | 32.00 | 88.83 | -1.95 | 86.52 |   85.85   |     **99.11**     |
>
> | Suboptimal traj. | # expert traj. |  BCE  |  BCU  | DWBC  |   CLARE   | DemoDICE | iLID   (ours) |
> | :--------------: | :------------: | :---: | :---: | :---: | :-------: | :------: | :-----------: |
> |      Random      |       1        | -0.32 | 2.25  | 0.89  |   -0.25   |   2.24   |   **2.43**    |
> |      Random      |       3        | 5.40  | 2.25  | 4.33  |   2.75    |   2.22   |   **5.77**    |
> |      Random      |       5        | 4.05  | 2.25  | 2.16  |   4.74    |   2.23   |   **5.86**    |
> |      Replay      |       1        | -0.32 | 23.56 | 9.15  |   31.07   |  30.96   |   **34.49**   |
> |      Replay      |       3        | 5.40  | 28.89 | 12.72 | **35.77** |  19.62   |     34.66     |
> |      Replay      |       5        | 4.05  | 35.10 | 19.33 |   37.05   |  28.25   |   **38.63**   |
> |      Medium      |       1        | -0.32 | 42.60 | 9.31  | **42.56** |  41.94   |     42.47     |
> |      Medium      |       3        | 5.40  | 42.86 | 5.90  |   42.48   |  39.88   |   **43.10**   |
> |      Medium      |       5        | 4.05  | 42.74 | 7.24  |   42.38   |  41.50   |   **43.38**   |
>
>
> **Q2: Why does BC with expert data (BCE) perform worse than BC with union data (BCU)?**
>
> BCE is trained only on the provided expert trajectories (in our experiments, BCE only uses 1 expert trajectory), whereas BCU utilizes both expert trajectories and diverse trajectories, resulting in differing sample sizes between the two methods. Hence, the performance of BCE is easily hampered by the limited data coverage of the expert dataset, leading to significant extrapolation errors.
>
> **Q3: Can the runtime for DemoDICE be included in Figure 5?**
>
> The reason we opted not to incorporate DemoDICE in Figure 5 stems from the divergence in frameworks. DemoDICE's official code is built upon the TensorFlow framework, while our work is grounded in PyTorch. This framework discrepancy makes the algorithms not directly comparable. We are actively working on reproducing DemoDICE using PyTorch and plan to integrate its result into forthcoming iterations of our work.

---

### Official Review · Reviewer_K4fk · 2023-07-05

**Soundness:** 3 good
**Presentation:** 3 good
**Contribution:** 3 good
**Rating:** 6
**Confidence:** 3

**Summary:**

The paper proposes an algorithm for offline imitation learning on a mixture of “perfect” expert demonstrations and “imperfect” sub-optimal demonstrations. The authors provide a theoretical motivation for their approach and exhibit results on various Mujoco and Adroit tasks. The authors also conduct ablation studies to justify their design choices.

**Strengths:**

- The paper proposes a novel and interesting approach for learning from suboptimal data. It also provides some theoretical motivation for why filtering data based on resultant states might be better than simply using the state-action distribution.
- The paper is written clearly and concisely.
- In addition to the novelty in filtering the data, the paper proposes a novel constrained BC algorithm where a discount factor is used to reduce the impact of stochasticity in the MDP during the optimization process.
- The author’s compare their approach with a bunch of recently published offline imitation learning algorithms capable of learning of suboptimal data. The algorithms are evaluated across 4 MoJoCo tasks and 4 Adroit tasks with two variations (low and high expert data) for each task.
- The authors justify their design choices using ablation studies showing the impact of number of expert demonstrations, the number of rollback steps considered, the use of the discount factor and a comparison between runtimes of iLID and other algorithms.


**Weaknesses:**

- Though the paper has some good ablation studies, it would be interesting to have a study of the effect of quality of the suboptimal data on the performance of the method. The paper currently only considers random trajectories as suboptimal data which might not be very useful for harder tasks. Instead, imagine demonstrations that can complete a part of the task but not the entire task (for instance, can pick up the hammer but not hit the nail). These can probably be collected by rolling out an expert agent and taking random actions in between (varying the percentage of random actions can give different levels of suboptimality). Such a study can help (1) highlight the relevance of the work in the real world where collecting perfect demos might be hard but it is often possible to do parts of the task, and (2) highlight the importance of expert demonstrations in this problem setting (for instance, can we reduce the amount of expert data if the amount of suboptimality in the remaining data reduces).
- Fig. 4 plots the average return against the number of training steps. However, since the models are completely trained on offline data, a better metric might be a comparison of maximum performance attained by different algorithms. Also, training it for too long might result in the model overfitting on the data, thus, reducing the average return over time (as can be seen in quite a few tasks in Fig. 4).
- It would be great if the ablations in Fig. 3 could be shown on a few more tasks in the appendix.
- The paper is missing a limitations section.


**Questions:**

It would be great if the authors could address the points mentioned in the “Weaknesses” section.

**Limitations:**

The paper is missing a limitations section.

---

> ### Author Rebuttal · Authors · 2023-08-09
>
> Thank you for the reviewer's appreciation of the contribution and novelty of this paper! Below are detailed responses to each comment:
>
> **Q1: It would be interesting to have a study of the effect of the quality of the suboptimal data on the performance of the method.**
>
> Following the reviewer's suggestion, we employ the Random, Replay, and Medium datasets in D4RL, which, ranging from low to high qualities (detailed below), serve as the suboptimal demonstrations (without seeding with expert data).
>
> > Random, Medium, and Replay use samples from 1) a randomly initialized policy, 2) a policy trained to approximately 1/3 the performance of the expert, and 3) the replay buffer of a policy trained up to the performance of the medium agent, respectively.
>
> We carry out a series of experiments in four MuJoCo environments with varying numbers of expert trajectories (from 1 to 5). The selected results are presented in the tables below (above: in Ant, below: in Halfcheetah; please refer to Figure 1 in the PDF for the complete set of results). The results demonstrate the efficacy of iLID in effectively extracting positive behaviors from suboptimal demonstrations across a spectrum of quality levels. As expected, the reduction in suboptimality corresponds to a decrease in the required number of expert trajectories (exemplified by the superior performance of 1 expert trajectory on the Replay data in Ant, compared to 3 expert trajectories on the Random Data). However, surprisingly, albeit with higher overall scores, the performance of iLID on the Medium data frequently falls short in comparison to its performance on the Replay data, revealing the larger state coverage and richer information embedded within the Replay data.
>
> | Suboptimal traj. | # expert traj. |  BCE  |  BCU  | DWBC  | CLARE | DemoDICE  | iLID   (ours) |
> | :--------------: | :------------: | :---: | :---: | :---: | :---: | :-------: | :-----------: |
> |      Random      |       1        | -2.37 | 31.55 | 2.86  | 31.73 |   32.03   |   **33.60**   |
> |      Random      |       3        | -2.18 | 31.48 | 23.67 | 17.71 | **49.63** |     45.13     |
> |      Random      |       5        | 32.00 | 31.48 | 41.46 | 31.75 |   39.48   |   **64.97**   |
> |      Replay      |       1        | -2.37 | 67.73 | 9.76  | 61.42 |   72.74   |   **80.20**   |
> |      Replay      |       3        | -2.18 | 69.82 | 30.11 | 64.33 |   79.97   |   **96.00**   |
> |      Replay      |       5        | 32.00 | 62.93 | 24.61 | 62.72 |   82.17   |   **97.65**   |
> |      Medium      |       1        | -2.37 | 87.67 | -0.08 | 86.24 | **92.07** |     82.96     |
> |      Medium      |       3        | -2.18 | 81.89 | 8.95  | 85.78 |   81.70   |   **89.33**   |
> |      Medium      |       5        | 32.00 | 88.83 | -1.95 | 86.52 |   85.85   |     **99.11**     |
>
> | Suboptimal traj. | # expert traj. |  BCE  |  BCU  | DWBC  |   CLARE   | DemoDICE | iLID   (ours) |
> | :--------------: | :------------: | :---: | :---: | :---: | :-------: | :------: | :-----------: |
> |      Random      |       1        | -0.32 | 2.25  | 0.89  |   -0.25   |   2.24   |   **2.43**    |
> |      Random      |       3        | 5.40  | 2.25  | 4.33  |   2.75    |   2.22   |   **5.77**    |
> |      Random      |       5        | 4.05  | 2.25  | 2.16  |   4.74    |   2.23   |   **5.86**    |
> |      Replay      |       1        | -0.32 | 23.56 | 9.15  |   31.07   |  30.96   |   **34.49**   |
> |      Replay      |       3        | 5.40  | 28.89 | 12.72 | **35.77** |  19.62   |     34.66     |
> |      Replay      |       5        | 4.05  | 35.10 | 19.33 |   37.05   |  28.25   |   **38.63**   |
> |      Medium      |       1        | -0.32 | 42.60 | 9.31  | **42.56** |  41.94   |     42.47     |
> |      Medium      |       3        | 5.40  | 42.86 | 5.90  |   42.48   |  39.88   |   **43.10**   |
> |      Medium      |       5        | 4.05  | 42.74 | 7.24  |   42.38   |  41.50   |   **43.38**   |
>
> **Q2: ...a better metric might be a comparison of maximum performance attained by different algorithms...**
>
> Thank you for pointing this out, we will include the information (the average of the highest 10 scores) in the appendix to assess the potential of the proposed algorithm. Combined with the average returns of the policy at the last iteration of training, we can provide a more comprehensive understanding of the algorithm's performance, in terms of both the stability and potential.
>
> **Q3: The ablations could be shown on a few more tasks in the appendix.**
>
> We conduct complementary ablation studies across four MuJoco environments. As showcased in Figure 3 within the attached PDF, the results further corroborate the significance of the constrained BC procedure. It not only accelerates training but also contributes to the proposed algorithm's stability.
>
> **Q4: The paper is missing a limitations section.**
>
> The main limitation of this paper lies in the requirement of the existence of state similarity between the suboptimal data and the expert data. However, it is worth noting that in the context of offline IL, when the states within suboptimal data deviate significantly from the given expert data, it is extremely challenging to assess the values of suboptimal behaviors with no prior information. Another limitation of this work is the lack of theoretical guarantees in the general MDPs, which we leave for future work.

---

> > ### Comment · Reviewer_K4fk · 2023-08-13
> > **Thank you for the rebuttal**
> >
> > I thank the authors for the rebuttal and additional experiments. All my concerns have been addressed and I am raising my score by a point.

---

> > > ### Author Response · Authors · 2023-08-14
> > > **Thank you**
> > >
> > > Thank you so much for further reviewing our response and increasing the score!

---

### Author Rebuttal · Authors · 2023-08-10

We sincerely thank the reviewers for the insightful and constructive feedback! We also thank the reviewers for their appreciation of the novelty (Reviewer K4fk), contribution (Reviewer Y7ma), and writing (Reviewers b1iR and 71T4) of our work. Per the reviewers' suggestions, we have: (1) carried out new experiments using suboptimal demonstrations that have different qualities and are not seeded with expert data, (2) added more ablation studies, (3) sharpened the motivating theoretical results by removing the assumption on the uniform distribution, and (4) replied to each reviewer's comment point by point. New comments on these responses are very welcome!

---

### Decision · Program_Chairs · 2023-09-21

**Decision:**

Reject

**Comment:**

While there are aspects of the submission here that strong, unfortunately, there was too much movement during the brief discussion period to be able to confidently declare that this paper is ready for NeurIPS. More specifically, there are some lingering doubts surrounding both the notation/correctness of Theorem 3.1 and also the nature/discussion of the algorithm’s performance when there is a large distribution shift between the expert and non-expert data. The authors are strongly encouraged to carefully consider the excellent feedback that they received from the reviewers on these and other points; a full revision designed to address this feedback should make for a strong submission to any future venue.